# Genome-Wide Identification and Comprehensive Analysis of the *AP2/ERF* Gene Family in Pomegranate Fruit Development and Postharvest Preservation

**DOI:** 10.3390/genes13050895

**Published:** 2022-05-17

**Authors:** Ran Wan, Jinhui Song, Zhenyang Lv, Xingcheng Qi, Xuemeng Han, Qiang Guo, Sa Wang, Jiangli Shi, Zaihai Jian, Qingxia Hu, Yanhui Chen

**Affiliations:** 1College of Horticulture, Henan Agricultural University, Zhengzhou 450002, China; wanxayl@henau.edu.cn (R.W.); sjh20120511@163.com (J.S.); 17637630353@163.com (Z.L.); 15294747067@163.com (X.Q.); 17515736581@163.com (X.H.); guoqiangair@163.com (Q.G.); 18137276468@163.com (S.W.); sjli30@henau.edu.cn (J.S.); jianzaihai@163.com (Z.J.); chenyanhui188@163.com (Y.C.); 2Henan Key Laboratory of Fruit and Cucurbit Biology, College of Horticulture, Henan Agricultural University, Zhengzhou 450002, China

**Keywords:** pomegranate, AP2/ERF gene family, gene expression, development, postharvest preservation

## Abstract

Pomegranate (*Punica granatum* L.) is a kind of fruit with significant economic, ecological and health values. AP2/ERF transcription factors belong to a large group of factors mainly found in plants and play key roles in plant growth and development. However, AP2/ERF genes in pomegranate and their implication in development and postharvest preservation have been little described. In this study, 116 *PgAP2/ERF* genes in pomegranate were identified and renamed based on their chromosomal distributions. Phylogenetic relationship with genes from other species, structures, duplications, annotations, *cis*-elements in promoter sequences, and protein-protein interaction networks among PgAP2/ERF proteins were comprehensively explored. Expression analysis revealed several *PgAP2/ERFs* associated with the phenotypes of pomegranate seed hardness, including *PgAP2/ERF5*, *PgAP2/ERF36*, *PgAP2/ERF58*, and *PgAP2/ERF86*. Subsequent analysis indicated that many differentially expressed *PgAP2/ERF* genes are potentially important regulators of pomegranate fruit development. Furthermore, expression of more than one-half of *PgAP2/ERFs* was repressed in ‘Tunisian soft seed’ pomegranate fruit under low-temperature cold storage. The results showed that 1-MCP implicated in promoting postharvest preservation of ‘Tunisian soft seed’ pomegranate upregulated the *PgAP2/ERF4*, *PgAP2/ERF15*, *PgAP2/ERF26*, *PgAP2/ERF30*, *PgAP2/ERF35* and *PgAP2/ERF45* genes compared to those under low-temperature cold storage. This indicates that these genes are important candidate genes involved in pomegranate postharvest preservation. In summary, the findings of the present study provide an important basis for characterizing the *PgAP2/ERF* family genes and provide information on the candidate genes involved in pomegranate fruit development and postharvest preservation.

## 1. Introduction

Pomegranates (*P. granatum*) are fruit that originated from Central Asia and are currently widely grown worldwide. Iran, India, and China contribute 80% of the global production of pomegranates [1]. Pomegranates have significant economic and ecological values and have attracted immense attention. They are widely consumed as fresh products and widely applied in industries for making dyestuffs, cosmetics, and medicines. Moreover, pomegranates have antibacterial, anticancer, heart protection, and anti-inflammatory activities [2,3,4].

Pomegranate fruit is classified as a non-climacteric fruit [5]. The ethylene-associated pathway is closely related to fruit development and postharvest processes of climacteric and non-climacteric fruit [6,7]. Ethylene responsive factors (ERFs), such as MdERF2 (*Malus domestica*), PpERFA16 (*Prunus persica*), MaERF9 (*Musa acuminata*), and MaERF11 (*M. acuminata*), are important components in the ethylene pathway and are implicated in regulating fruit ripening by modulating ethylene biosynthesis [8,9,10,11]. ERFs are also implicated in fruit postharvest preservation. Significant differential expression of *MdERFs* was reported in long-term low-temperature condition storage of apple fruit [12] and *MdERF2* have the function of improving apple preservation through delaying ripening [10]. EjERF39 (*Eribotrya japonica*) can form a complex with EjMYB8, induce loquat fruit lignification at low-temperature storage, and increase chilling injury [13].

Pomegranate fruit are easily affected by decay and chilling injury during postharvest cold storage, thus decreasing the fruit quality [6,14]. Comparative transcriptomics of the relatively chilling-tolerant ‘Wonderful’ and the relatively chilling-sensitive ‘Ganesh’ pomegranate fruit under cold storage revealed some differentially expressed *ERF*, *DREB*, and *RAP* genes [15,16,17,18]. These results highlight the implication of *PgAP2/ERFs* in pomegranate postharvest preservation. 1-MCP (1-methylcyclopropene) improves the preservation effects of postharvest fruit by competitively binding with ethylene receptors, which bind with ethylene to activate the ethylene-associated pathway [19]. Exposure to 1-MCP decreases peel browning and maintains the internal quality of different pomegranate varieties [14,20]. The implication of *PgAP2/ERFs* in 1-MCP pomegranate postharvest preservation remains unrevealed so far.

ERF is a subfamily of the AP2/ERF (APETALA2/ethylene-responsive factor) super family in plants, which is involved in regulation of diverse plant processes [21,22]. Members of this super family are characterized by at least one highly conserved AP2 DNA-binding domain, comprising 60–70 conserved amino acids [23]. The AP2/ERF super-family factors are mainly classified into four major subfamilies, namely, ERF (Dehydration-Responsive Element-Binding and Ethylene-Responsive Element-Binding protein, having only one AP2 domain), AP2 (APETALA2, having two AP2 domain repeats), RAV (Related to ABI3/VP, having an AP2 domain and a specific B3 DNA-binding domain), and Soloists (few unclassified factors, having only one AP2 domain) [21,22,24]. The AP2/ERF super-family has been identified and explored in several herbaceous and ligneous plants, including *Arabidopsis thaliana* having 147 members, rice (*Oryza sativa*) having 164 members, grapevine (*Vitis vinifera*) having 149 members, pear (*Pyrus bretschneideri*) having 155 members, longan (*Dimocarpus longan*) having 125 members, and pineapple (*Ananas comosus* L. Merr) having 97 members [21,25,26,27,28].

The implication of the *AP2/ERF* genes in pomegranate fruit development and postharvest preservation has not been fully elucidated. In consideration of the limited information available on pomegranate *AP2/ERF* genes, in the present study, pomegranate genes of the AP2/ERF family were identified, and their structures, conserved domains, and phylogenetic relationships were systemically analyzed. Furthermore, the expression profiles of the *PgAP2/ERF* genes in different pomegranate tissues as well as during pomegranate fruit development and postharvest preservation were explored. Our results provide key information on the pomegranate *AP2/ERF* genes and significant insights into the role of *PgAP2/ERF* genes in pomegranate fruit development and postharvest preservation.

## 2. Materials and Methods

### 2.1. Plant Materials, Browning Index, and Ethylene Production

‘Tunisian soft-seed’ pomegranate fruit were harvested at commercial maturity from the pomegranate orchards at Liugou village, Gaocun town, Xingyang city, Henan province, China, on 10 October 2020. The fruit were used for postharvest preservation experiments with three biological replicates. Parts of the fruit were marked ‘LTC’ to show that the fruit was subjected to cold storage treatment at a low temperature of 4 °C. The ‘1-MCP + LTC’ groups comprised fruits with 1-MCP patches under the ‘LTC’ treatment. The effective concentration of 1-MCP was 1.0 μL/L. Fruit peels were collected at 0, 15, 30, 45, and 60 days post treatment and photos of the pomegranate fruit were taken. The samples were then immediately frozen in liquid nitrogen and stored at −80 °C for subsequent gene expression analysis.

Pomegranate fruit under LTC and 1-MCP + LTC were graded into four classes based on the peel-browning percentages as follows: 0 (no browning symptom and with the percentage of 0), 1 (1–25%), 2 (26–50%), 3 (51–75%), and 4 (>75%). The browning index was calculated as follows: Browning index = ∑(browning grade × number of browning fruit)/5 × total number of fruit [29].

Ethylene production during the postharvest preservation experiment was evaluated by gas chromatography (GC 2010PLUS, Shimadzu, Kyoto, Japan) [30]. Two fruit were placed in a 2 L box sealed with an air-tight lid equipped with a rubber stopper at room temperature for three hours to collect the ethylene gas released by the fruit. Three boxes were used for the three technical replicates for each biological replicate. Subsequently, 1 mL of gas was obtained from each box and subjected to GC analysis with three replicates using helium as a carrier gas (35 mL min^−1^), hydrogen (40 mL min^−1^), and air (350 mL min^−1^). Injector and detector temperatures were set at 110 and 200 °C, and a 10 μL L^−1^ ethylene standard was used for equipment calibration.

### 2.2. Identification and Annotation of AP2/ERF Genes in the Pomegranate Genome

The ‘Tunisian soft-seed’ pomegranate genome was retrieved from the NCBI Sequence Read Archive database (https://www.ncbi.nlm.nih.gov/assebly/GCF_007655135.1 accessed on 16 March 2021). Protein sequences of the 147 AtAP2/ERFs were obtained from TAIR webserver (http://www.arabidopsis.org/ accessed on 16 March 2021). HMMER and BLASTP tools were used to screen the candidate PgAP2/ERF genes according to the AP2 domain model retrieved from Pfam database and AtAP2/ERF protein sequences. NCBI CDD (Conserved Domain Database; https://www.ncbi.nlm.nih.gov/Structure/bwrpsb/bwrpsb.cgi accessed on 18 March 2021) webserver and family assignment algorithms were used to identify the PgAP2/ERF genes. The identified PgAP2/ERF genes were renamed according to their positions on the pomegranate chromosomes. Isoelectric points (pI) and molecular weights (MW) of the PgAP2/ERF proteins were predicted using the ExPASy proteomic webserver (https://web.expasy.org/compute_pi/ accessed on 18 March 2021).

### 2.3. Analysis of Phylogenetic Relationships, Conserved Motifs, Gene Structures and Duplications of PgAP2/ERF Genes

Phylogenetic trees were built based on full-length protein sequences of *PgAP2/ERFs* by the maximum likelihood (ML) method with 1000 bootstrap values using MEGA software (version 7.0, Mega Limited, Auckland, New Zealand). Conserved motifs in *PgAP2/ERF* genes were analyzed using the MEME webserver (https://meme-suite.org/meme/tools/meme accessed on 15 May 2021). Exon-intron structures and chromosomal locations of *PgAP2/ERF* genes were visualized using TBtools (SCAU) [31]. Gene duplications of *PgAP2/ERFs* were analyzed using the Multiple Collinearity Scan tool kit in TBtools with the default parameters [31]. Duplicate segments of the *PgAP2/ERFs* of pomegranate and other plants were then selected using the Dual Systeny Plotter function in TBtools [31]. Ratios of nonsynonymous substitution (Ka) to synonymous substitution (Ks) of *PgAP2/ERF* orthologous gene pairs were calculated using TBtools [31].

### 2.4. Gene Ontology, Cis-Elements, and Protein-Protein Interaction Network Analysis of the PgAP2/ERF Genes

Gene ontology (GO) analysis of the *PgAP2/ERF* genes was performed using the PANNZER2 webserver (Protein ANNotation with Z-scoRE http://ekhidna2.biocenter.helsinki.fi/sanspanz/ accessed on 25 May 2021). For sequences upstream, 2000 bp of the *PgAP2/ERF* gene sequences were submitted to the PlantCARE tool (http://bioinformatics.psb.ugent.be/webtools/plantcare/html/ accessed on 25 May 2021) to explore the *cis*-elements present in the promoter regions of the *PgAP2/ERFs*. The results were then visualized using TBtools [31]. Homologues of the *PgAP2/ERF* genes in *Arabidopsis* were obtained to explore the interacting proteins for the protein-protein interaction network (PPI) analysis using the STRING webserver (http://string-db.org accessed on 25 May 2021).

### 2.5. Expression Analysis of PgAP2/ERF Genes during Pomegranate Development and Postharvest Preservation

RNA-seq data (Accession Numbers are presented in Appendix A) from six different pomegranate tissues and organs were retrieved from NCBI Sequence Read Archive database (http://www.ncbi.nlm.nih.gov/sra accessed on 16 May 2021). The samples included roots, fresh leaves, flowers, peels, inner seed coats, and outer seed coats at different development stages. Transcripts per kilobase values (TPM) of the exon model per million mapped reads was calculated using Trimmomatic [32] and Kallisto tools [33]. *PgAP2/ERF* expression profiles of pomegranate peels under LTC were obtained from the corresponding transcriptomic data (unpublished). Heat maps were constructed using the HeatMap function in TBtools [31]. Normalized expression levels were obtained for comparison across treatments.

Primers were designed using Primer 5 software (http://www.premierbiosoft.com/primerdesign/, accessed on 15 April 2021) and are presented in Appendix A. RNA extraction, DNase treatment, and reverse transcription were performed as previously described [34]. qRT-PCR analyses were performed using SYBR Green qPCR Master Mix (Vazyme, Nanjing, China) on a CFX96 Real Time System (Bio-Rad, Hercules, CA, USA) with three biological replicates and three technical replicates. The qRT-PCR conditions were as follows: 95 °C for 5 min, followed by 40 cycles of 95 °C for 15 s and 60 °C for 1 min, and 72 °C for 5 min. Relative gene expression levels were calculated using the 2^−ΔΔCt^ method [35].

## 3. Results

### 3.1. Identification of the PgAP2/ERF Genes in Pomegranate

A total of 116 members of the *PgAP2/ERF* gene family were retrieved from the ‘Tunisian soft-seed’ pomegranate genome and were renamed from *PgAP2/ERF1* to *PgAP2/ERF11**6* according to their positions on the chromosomes (Appendix A). The length of the proteins encoded by these genes ranged from 130 to 612 amino acids, and the molecular weights of the proteins ranged from 14,692.26 to 84,711.64 Da (Appendix A). The theoretical isoelectric points of these proteins ranged from 4.57 to 11.65, with PgAP2/ERF79 having the lowest isoelectric point of 4.57 whereas PgAP2/ERF102 had the highest isoelectric point (11.65). The proteins comprised 79 acidic proteins and 38 basic proteins (Appendix A). Prediction of subcellular location showed that 77 out the 116 PgAP2/ERF proteins were localized in the nucleus whereas the other proteins were localized in the nucleus and cytoplasm, except for the PgAP2/ERF64 protein that was localized in the cytoplasm (Appendix A). These results provide a theoretical basis for further studies on the purification, activity, and function of PgAP2/ERF proteins.

### 3.2. Phylogenetic Analysis and Classification of PgAP2/ERF Genes

Phylogenetic analysis showed that the *PgAP2/ERF* gene family comprised four subfamilies, including 102 *PgERF* genes, 10 *PgAP2* genes, 3 *PgRAV* genes, and one *PgSoloist* gene (Figure 1). The number of *PgERF* subfamily genes accounted for 88% of the entire gene family and there were seven genes encoding C-repeat binding factors (CBFs) (Figure 1; Appendix A), which has been reported previously [36]. The *ERF* subfamily was further divided into 12 ERF groups according to the phylogenetic results (Figure 1). In fact, 11 ERF groups were identified because ERF Ⅺ had no pomegranate analog (Figure 1). The ERF Group Ⅴ had a relatively distant genetic relationship from the other subfamilies, forming an independent branch (Figure 1). The ERF Group Ⅷ comprised twenty *PgERFs*, the highest number in all the ERF groups. The ERF Group Ⅻ comprised only two gene members, which was the lowest number among all the ERF groups (Figure 1).

### 3.3. Conserved Motifs and Gene Structures of PgAP2/ERF Genes

Conserved motifs and gene structures of the *PgAP2/ERF* family genes were explored using the MEME webserver (Figure 2; Appendix A). The results showed that genes clustered on adjacent branches on the phylogenetic tree of the same subfamily exhibited highly conserved motifs (Figure 2a). Each *PgAP2/ERF* gene showed motif 1 and 3, except the Soloist subfamily gene *PgAP2/ERF114* only having motif 3, and motifs 4–10 were differentially distributed in different *Pg**AP2/ERF* genes, indicating the variability in this gene family. Moreover, the AP2 domain was conserved in all *PgAP2/ERF* genes (Figure 2a). In addition, 80 *PgERFs* and 3 *PgRAVs* showed one intron, and the other 17 *PgERFs* had two introns, whereas 10 *PgAP2* and 1 *PgSoloist* had at least six introns and the *PgAP2/ERF47* gene had the greatest number of introns (9) (Figure 2b). These findings showed a relatively high degree of conservation in the *PgAP2/ERF* gene family.

### 3.4. Chromosomal Locations and Gene Duplications of the Pomegranate PgAP2/ERF Genes

PgAP2/ERF genes were differentially located on eight pomegranate chromosomes (LG1-8) except for *PgAP2/ERF116* located on Contig 00460 (Figure 3a; Appendix A). *PgAP2/ERFs* were mostly distributed at the two ends of the corresponding chromosomes (Figure 3a), where they always exhibited high gene densities [37]. Chromosome LG4 had 22 *PgAP2/ERF* genes, LG1 had 21 genes, LG3 had 18 genes, LG2 had 14 genes, and LG7 had the lowest number, 8 *PgAP2/ERFs* (Figure 3a; Appendix A). The results showed no positive correlation between chromosome length and number of the *PgAP2/ERF* genes.

Gene collinearity analysis of the *PgAP2/ERF* gene family showed three tandem duplication events and 43 segmental duplication events (indicated by homologous gene pairs), involving 68 *PgAP2/ERFs*, accounting for 59% members of the whole gene family (Figure 3b; Appendix A). The Ka/Ks ratio of the homologous gene pair can be used as an indicator of selection pressure in the process of gene evolution. Ka/Ks ratios less than 1.0 indicate a lower selection pressure [38]. The six homologous gene pairs in the present study displayed Ka/Ks ratios above 1.0, and 37 homologous gene pairs exhibited ratios below 1.0 (Appendix A), indicating that these genes underwent the purification selection pressure.

Syntenic analysis of the PgAP2/ERF genes and AP2/ERFs from dicotyledonous plants, Arabidopsis and grape, as well as from monocotyledonous plants, rice and maize, were further conducted (Figure 4). The results revealed that 127, 127, 53, and 52 syntenic gene pairs were present between the AP2/ERF genes in pomegranate and those in Arabidopsis, grape, maize, and rice, respectively. Among them, 21 common syntenic gene pairs were observed (Figure 4; Appendix A), and it indicated that these genes may have existed before the differentiation of monocotyledons and dicotyledons.

### 3.5. Cis-Elements in the Promoters of PgAP2/ERF Genes

*Cis*-elements of all the promoter sequences, 2000 bp upstream of the *PgAP2/ERF* coding sequences, were predicted using the PlantCARE webserver (Figure 5a). The *cis*-elements of the *PgAP2/ERF* promoters can be generally assigned to four categories: light-responsive elements, hormone-responsive elements, stress-responsive elements as well as growth- and development-responsive elements (Figure 5b). Promoters of all the *PgAP2/ERFs* were characterized by light-responsive elements, such as G-box and Box4, and were also characterized by hormone-responsive elements (Figure 5). A total of 106 *PgAP2/ERF* promoters exhibited the abscisic acid (ABA) response element ABRE, 98 *PgAP2/ERF* promoters were characterized by two methyl jasmonate (MEJA)-response elements, 43 genes exhibited the gibberellin (GA)-response element P-box, 39 *PgAP2/ERF* promoters showed the presence of the ethylene (ETH)-response element ERE, and 36 *PgAP2/ERF* promoters had the auxin (AUX)-response element TGA (Figure 5). Moreover, all *PgAP2/ERF* promoters exhibited stress-responsive elements, including the anaerobic ARE, the low-temperature responsive LTR, the drought-inductive MBS, as well as the defense- and stress-responsive TC-rich repeats and WUN (Figure 5). Growth and development elements were observed in the promoters of 63 *PgAP2/ERFs*. RY-elements involved in seed-specific regulation were observed in 14 *PgAP2/ERF* promoters and HD-Zip1 elements, involved in the differentiation of palisade mesophyll cells, were present in 7 *PgAP2/ERF* promoters (Figure 5).

### 3.6. Gene Ontology and Protein-Protein Interaction Network of PgAP2/ERF Genes

To further explore the possible biological functions of the *PgAP2/ERF*s, gene annotation was performed through gene ontology (GO) analysis (Figure 6; Appendix A). GO analysis of the biological processes category showed that *PgAP2/ERF* genes participated in multiple biological processes, and the top five terms were ‘regulation of transcription, DNA-templated’ (GO:0006355), ‘defense response’ (GO:0006852), ‘ethylene-activated signaling pathway’ (GO:0009873), ‘response to chitin’ (GO:0010200), and ‘positive regulation of nucleic acid-templated transcription’ (GO:1903508) (Figure 6). All *PgAP2/ERF*s except *PgAP2/ERF26* were associated with the GO term ‘regulation of transcription and DNA-templated’ (Figure 6; Appendix A). GO analysis of the molecular function category showed that all *PgAP2/ERF* family genes were associated with ‘DNA-binding transcription factor activity’ (GO:0003700), ‘transcription regulatory region nucleic acid binding’ (GO:0001067), and ‘protein binding’ (GO:0005515) GO terms (Figure 6). Cellular component analysis revealed that most of the *PgAP2/ERF* genes were localized in the nucleus, whereas some *PgAP2/ERF* genes, such as *PgAP2/ERF113* and *PgAP2/ERF54*, were localized on the integral membrane component and in cytoplasm (Figure 6; Appendix A).

Furthermore, the PPI network of PgAP2/ERFs was preliminarily predicted based on *Arabidopsis* homologous proteins, and a complex PPI network of the PgAP2/ERFs and AtERFs was established (Figure 7). AtRRTF1 (PgAP2/ERF41, PgAP2/ERF67, PgAP2/ERF27, and PgAP2/ERF28) had potential interaction relationship with AtORA47 (PgAP2/ERF108), and AtERF13 (PgAP2/ERF57, PgAP2/ERF62 and PgAP2/ERF63) as well as with AtWRI1 (PgAP2/ERF106) (Figure 7). AtORA47 (PgAP2/ERF108) exhibited possible interaction with AtERF13 (PgAP2/ERF57, PgAP2/ERF62 and PgAP2/ERF63) and AtERF4 (PgAP2/ERF4, PgAP2/ERF30, PgAP2/ERF86, and PgAP2/ERF105) (Figure 7). In addition to the predictions above, AtERF5 (PgAP2/ERF55 and PgAP2/ERF14) showed a strong interaction relationship with AtORA47 (PgAP2/ERF108), AT5G51190 (PgAP2/ERF35), AtERF13 (PgAP2/ERF57, PgAP2/ERF62, and PgAP2/ERF63) and AtERF4 (PgAP2/ERF4, PgAP2/ERF30, PgAP2/ERF86, and PgAP2/ERF105) (Figure 7). These results provided a preliminary reference for further study of the interactions between pomegranate AP2/ERFs.

### 3.7. Tissue-Specific Expression of PgAP2/ERF Genes

Transcriptomic data of different tissues (leaf, root, flower, peel, inner, and outer seed coats) of ‘Dabenzi’, a hard-seeded pomegranate cultivar, and ‘Tunisian soft-seed’, a soft-seeded pomegranate cultivar, were retrieved from the NCBI SRA webserver for tissue-specific expression analysis of *PgAP2/ERF* genes (Figure 8). A high variance in expression levels of the *PgAP2/ERF* genes was observed across the investigated tissues (Figure 8). *PgAP2/ERF20* and *PgAP2/ERF42* genes were specifically expressed in pomegranate leaves; *PgAP2/ERF18*, *PgAP2/ERF112*, and *PgAP2/ERF64* were specifically expressed in pomegranate roots; *PgAP2/ERF45* and *PgAP2/ERF59* were specifically expressed in pomegranate flowers; and the *PgAP2/ERF8* gene was mainly expressed in pomegranate peels (Figure 8). In addition, *PgAP2/ERF23* and *PgAP2/ERF107* were specifically expressed in pomegranate inner seed coats; and *PgAP2/ERF75* and *PgAP2/ERF89* were specifically expressed in pomegranate outer seed coats (Figure 8). The number of *PgAP2/ERF* genes highly expressed in pomegranate roots was higher relative to that in the other investigated pomegranate tissues. The number of *PgAP2/ERF* genes expressed in inner and outer seed coats of ‘Tunisian soft-seed’ was higher compared with that expressed in inner and outer seed coats of ‘Dabenzi’. Differential expression of *PgAP2/ERF* genes was observed between the seed coats of ‘Dabenzi’ and ‘Tunisian soft-seed’ pomegranates. For example, *PgAP2/ERF5*, *PgAP2/ERF36*, *PgAP2/ERF88*, and *PgAP2/ERF97* were specifically expressed in ‘Dabenzi’ seed coats, whereas *PgAP2/ERF99*, *PgAP2/ERF58*, *PgAP2/ERF86*, and *PgAP2/ERF87* were only expressed in ‘Tunisian soft-seed’ seed coats (Figure 8).

### 3.8. Expression Profile of PgAP2/ERF Genes during Pomegranate Fruit Development

Transcriptomic data of the inner and outer seed coats of ‘Tunisian soft-seed’ pomegranate at 50, 95, and 140 days after flowering (DAF) and transcriptomic data of the peels of ‘Taishanhong’ pomegranate at 60, 90, and 150 DAF (Figure 9) were retrieved from the NCBI SRA database to explore the characteristics of *PgAP2/ERFs* during pomegranate fruit development. Notably, the whole seeds at 50 DAF were used towing to the challenge in visually distinguishing the inner and outer seed coats [39].

*PgAP2/ERFs* were assigned to six groups according to their expression profiles in inner and outer seed coats of pomegranate (Figure 9a). Most of the genes from Groups Ⅰ and Ⅱ were upregulated during pomegranate fruit development, and higher expression levels were observed in the inner seed coats than in the outer seed coats, especially at 140 DAF (Figure 9a). On the contrary, most genes from Group Ⅲ exhibited higher expression levels in the outer seed coats than in the inner seed coats mainly at 140 DAF (Figure 9a). Genes from Groups Ⅳ, Ⅴ, and Ⅵ showed relatively higher expression levels in the outer seed coats at 95 DAF, in the inner seed coats at 95 DAF stage, and the whole seeds at 40 DAF, respectively. *PgAP2/ERF1* and *PgAP2/ERF12* genes were not expressed in all seed coat samples (Figure 9a).

*PgAP2/ERFs* were assigned to four groups according to their expression patterns in peels during pomegranate development (Figure 9b). Expressions of all genes from group Ⅰ were downregulated during peel development, whereas expressions of all genes from Group Ⅱ were upregulated from 60 DAF to 90 DAF and downregulated from 90 DAF to 150 DAF (Figure 9b). Group Ⅲ comprised 15 genes mainly upregulated in peels at 150 DAF and 12 genes that were not expressed during peel development. Genes in Group Ⅳ showed relatively higher expression levels at 60 DAF relative to the expression levels at the other stages (Figure 9b).

### 3.9. Expression Profiles of PgAP2/ERF Genes during Pomegranate Postharvest Preservation

RNA-Seq data of the ‘Tunisian soft-seed’ pomegranate fruit during LTC were obtained (unpublished) to explore the possible roles of the *PgAP2/ERF* genes in postharvest preservation of pomegranate fruit. The results revealed six different expression clusters and important roles of *PgAP2/ERFs* during postharvest preservation (Figure 10). Group Ⅰ included 22 genes such as *PgAP2/ERF3*, *PgAP2/ERF109*, *PgAP2/ERF15*, *PgAP2/ERF2*5 and *PgAP2/ERF35*, which were upregulated at 15 DAT (days after the treatment) and downregulated at 45 DAT. Group Ⅵ comprised 9 genes that were downregulated at 15 DAT and relatively upregulated at 45 DAT, whereas *PgAP2/ERF88*, *PgAP2/ERF110*, *PgAP2/ERF22*, and *PgAP2/ERF32* in group Ⅱ were relatively upregulated after LTC (Figure 10). Group Ⅲ comprised 42 *PgAP2/ERF*s, which were directly repressed at 15 DAT, including *PgAP2/ERF4*, *PgAP2/ERF113*, *PgAP2/ERF8*, *PgAP2/ERF105*, *PgAP2/ERF26*, *PgAP2/ERF30*, and *PgAP2/ERF58*, etc. (Figure 10). In addition, 14 genes in Group Ⅳ were downregulated during the LTC period and 15 genes in Group Ⅴ were not upregulated until 45 DAT (Figure 10).

Postharvest application of 1-MCP is an efficient strategy used for the prevention of fruit damage under low-temperature storage [19]. Therefore, the preservation effect of 1-MCP under LTC (1-MCP + LTC) on the postharvest ‘Tunisian soft-seed’ pomegranate fruit was explored in the present study (Figure 11). Time series of fruit were sampled to further evaluate the relationship between *PgAP2/ERFs* and postharvest preservation (Figure 11 and Figure 12). The LTC fruit showed extensive browning appearance with significantly higher browning indexes compared with the 1-MCP + LTC fruit from 30 DAT (Figure 11a,b), indicating a positive effect of 1-MCP on improving the exterior quality of the postharvest pomegranate fruit. The ethylene production level of the 1-MCP + LTC fruit was also significantly lower relative to the LTC fruit at 30 and 60 DAT (Figure 11c).

Further, we screened the 12 genes shown in Figure 12 from Group Ⅲ as well as *PgAP2/ERF34* from Group Ⅳ, in which the genes mainly were downregulated after LTC. Compared to fruit at 0 DAT, the expression of *PgAP2/ERF4*, *PgAP2/ERF8*, *PgAP2/ERF15*, *PgAP2/ERF26*, *PgAP2/ERF34*, and *PgAP2/ERF35* was directly downregulated at 15 DAT and then slightly upregulated from 30 DAT under LTC. However, *PgAP2/ERF4*, *PgAP2/ERF15*, *PgAP2/ERF26*, and *PgAP2/ERF35* were significantly upregulated by the application of 1-MCP under LTC, especially from 15 to 30 DAT, despite their expression levels being lower than those at 0 DAT. Then, these three genes were downregulated after 45 DAT but almost with no significance between LTC and 1-MCP + LTC. On the contrary, *PgAP2/ERF8* and *PgAP2/ERF34* were more downregulated by the application of 1-MCP under LTC, especially from 15 to 30 DAT, compared to those under LTC. Afterwards, the two genes expressed with no significance between LTC and 1-MCP + LTC (Figure 12). There were also *PgAP2/ERF30* and *PgAP2/ERF 58* that were significantly upregulated by the application of 1-MCP under LTC, especially from 15 to 30 DAT, but the two genes expressed significantly lower from 45 DAT than those under LTC. The expression levels of *PgAP2/ERF45* were significantly higher under 1-MCP + LTC than those under LTC. *PgAP2/ERF25*, *PgAP2/ERF105*, and *PgAP2/ERF113* were not upregulated by the application of 1-MCP under LTC except for those at 15 DAT, and they mostly expressed significantly more highly under LTC than 1-MCP + LTC (Figure 12).

## 4. Discussion

*AP2/ERF* gene family is a major plant transcriptional factor family and is implicated in various developmental processes in plants [22]. The *AP2/ERF* gene family has been reported in several fruit trees, such as grapes, Chinese jujube, sugarcane, Chinese lotus, and pineapple [26,27,40,41,42]. However, studies have not explored the presence and function of the *AP2/ERF* gene family in pomegranate. Pomegranate fruit exhibits excellent nutritional qualities and health-promoting phytochemicals. Therefore, the present study was conducted to explore detailed gene information of pomegranate and evaluate the role of genes in the *AP2/ERF* gene family in pomegranate fruit development and postharvest preservation.

In total, 116 members of the *PgAP2/ERF* gene family were identified in the present study (Figure 1), based on the genomic information of ‘Tunisian soft-seed’ pomegranate available in public databases [37]. The number of *AP2/ERF* gene family members depends on the number of ERF subfamily members [43]. In the current study, the numbers of *PgAP2/ERF* gene family and ERF subfamily members (116, 102, *P. granatum*) were not significantly different relative to those of *Arabidopsis* (147, 122) and rice (164, 139 *O. sativa*) [21], but they were both less compared with the number of genes in cotton (269, 222 *Gossypium raimondii*) and celery (206, 172 *Apium graveolens*) [44,45]. Previous studies have reported that most genes in the same cluster present similar conserved motifs and exon-intron patterns [26]. The findings of the present study indicated that *PgAP2/ERF*s in the same *PgAP2/ERF* subfamily exhibited highly conserved motifs and gene structures. Notably, most *PgAP2/ERF*s similarly processed motif1, motif3, and AP2 domain (Figure 2), further indicating high conservation in the same subfamily as well as in the whole gene family.

Gene duplication events occurring during genome evolution are the main factors that lead to gene expansion and evolution, and they play important roles in improving plant adaptability to various environmental stresses [46,47,48]. In the present study, tandem repeat events and segmental duplication events were identified in pomegranate the *Pg**AP2/ERF* gene family, and syntenic pairs observed in *AP2/ERF* genes of pomegranate compared with those in *Arabidopsis*, grape, maize, and rice (Figure 3 and Figure 4). These results showed that gene duplication events were also key factors in the evolution and expansion of the *PgAP2/ERF* gene family.

Promoter *cis*-elements play key roles in regulation of gene expression [49]. The presence of different *cis*-elements in promoters of *Pg**AP2/ERF* genes indicates that they are involved in different regulatory networks related to ETH-, ABA-, and MEJA-associated pathways and responses to stresses (Figure 5). Similar results have been reported for *AP2/ERF* genes in other plant species [26,41,50]. Seventy *PgAP2/ERFs* containing LTR elements were identified, which included 7 *PgCBFs* (Figure 5, Appendix A), implying their potential roles in the response of pomegranates to cold. *CBF* genes belong to the *AP2/ERF* gene family and bind to a *cis*-element (DRE/CRT/LTRE) with a conserved CCGA core sequence, implicated in inducing expression of genes associated with increased cold tolerance [51,52]. Moreover, some transcription factors in the AP2/ERF family are modulated by other AP2/ERF-type factors to regulate the physiological processes of plants. For example, CBF2 negatively regulates the expression of CBF1 and CBF3, thus decreasing plant tolerance to multiple stresses [22]. PPI analysis in this study also showed some possible complex regulation interactions among the different PgAP2/ERFs but these PgAP2/ERFs include no PgCBF (Figure 7).

Transcriptomic analysis is usually used to study the mRNA expression levels of specific tissue or cells transcribed during a certain period, and to explore the related genes and phenotypes [53]. Transcriptomic analysis in the present study indicated *PgAP2/ERF5, PgAP2/ERF36, PgAP2/ERF88*, and *PgAP2/ERF97* exhibited high expression levels in seed coats of hard-seeded ‘Dabenzi’ pomegranate, whereas *PgAP2/ERF99, PgAP2/ERF58, PgAP2/ERF86*, and *PgAP2/ERF87* showed high expression levels in seed coats of soft-seeded ‘Tunisian soft seed’ pomegranate (Figure 8). In fact, *AP2*-like genes have been reportedly implicated in modulating seed hardness as these genes exhibit genetic divergences in soft-and hard-seeded pomegranate cultivars [37]. The differential expression of these genes above were likely related to the different phenotypes of soft-seeded and hard-seeded pomegranates, which is an important trait of pomegranate fruit that determines consumer preference.

Differentially expressed *AP2/ERF* genes are implicated in fruit development during the ripening process of several kinds of fruit and vegetable plants, such as grape (*V. Vinifera*), banana (*M. acuminata*), peach (*P. persica*), and tomato (*Solanum lycopersicum*) [11,54,55,56]. Many mechanisms of ERF regulating fruit development have been well exposed. MaERF11 is a potential repressor, whereas MaERF9 is an activator of the *MaACO1* gene family during banana fruit development by regulating cell expansion [9] and *PpERF3* regulates *PpNCED2/3* expression to promote ABA biosynthesis during fruit ripening in peach [57]. The results in the current study showed that *PgAP2/ERFs* with different expression levels are importantly involved in pomegranate fruit development (Figure 9), which deserved investigation of the possible roles of *PgAP2/ERFs* in pomegranate fruit development.

Cold storage under low temperature is widely used to maintain postharvest fruit quality, including pomegranate fruit [18]. Previous studies reported several differentially expressed *PgAP2/ERFs* by comparing the transcriptomic responses of the ‘Wonderful’ pomegranate fruit under three pairwise postharvest preservation treatments [15,16,17,18]. AP2/ERFs have proven to be important regulators of the preservation of different fruit, such as grapes, apples, loquats, and bananas, under LTC [11,12,13,56]. In ‘Tunisian soft-seed’ pomegranate fruit peels, lots of *PgAP2/ERFs* were found differentially expressed during LTC (Figure 10). It has been known that 1-MCP can keep postharvest storage performances of non-climacteric pomegranate fruit by modulating decaying scald incidence, peel browning, ethylene production, and other physical processes [6,19,20]. The present study also proved that 1-MCP can help the postharvest preservation of ‘Tunisian soft-seed’ pomegranate fruit under LTC through decreasing peel browning and ethylene production (Figure 11). Here, PgAP2/ERFs were also found importantly involved in the postharvest preservation of ‘Tunisian soft-seed’ pomegranate fruit by the application of 1-MCP under LTC. For example, *PgAP2/ERF4*, *PgAP2/ERF15*, *PgAP2/ERF26*, *PgAP2/ERF30*, *PgAP2/ERF35*, and *PgAP2/ERF58* were significantly upregulated by the application of 1-MCP under LTC, which were downregulated after LTC (Figure 12), which indicated their possible roles in pomegranate postharvest preservation.

PgAP2/ERF4 was implicated in the ‘negative regulation of ethylene-activated signaling pathway’ GO term (Figure 6; Appendix A) and is a homologue of MdERF4 (Figure 1). MdERF4 was proven to be involved in negatively regulating ethylene-mediated salt tolerance [58]. Therefore, we suggested that PgAP2/ERF4 might act as an important candidate regulator implicated in negatively regulating the ethylene-associated pathway in the effects of 1-MCP on the postharvest preservation of ‘Tunisian soft-seed’ pomegranate fruit. *PgAP2/ERF15* was identified as a CBF gene and was homologous to *AtCBF1* (Figure 1; Appendix A), which plays important roles in plant cold tolerance [59,60]. Recent studies have explored the roles of CBFs in fruit postharvest preservation during cold storage. For example, *MiCBF1* is induced in mango fruit under hot water treatment; *CmCBF1* and *CmCBF3* are induced in Hami melon fruit under LTC; and the *AcCBF* gene is induced in kiwifruit fruit under nitric oxide treatment. They all contribute to enhancing the cold tolerance of fruit to improve postharvest preservation [61,62,63]. Moreover, PgAP2/ERF30 is homologous to MaERF11, and MaERF11 is a crucial transcriptional repressor of ethylene biosynthesis in decaying banana fruit and postharvest ripening [9,64]. These findings indicate that *PgAP2/ERF15* and *PgAP2/ERF30* are also important candidate genes in pomegranate postharvest preservation.

## 5. Conclusions

A comprehensive and systematic genome-wide identification and analysis of *AP2/ERF* genes in the pomegranate genome was performed using bioinformatics tools and algorithms. A total of 116 *PgAP2/ERF* genes were identified and classified into four subfamilies with detailed information of the chromosomal distribution, phylogenetic analysis, gene structures, duplications, and their possible biological functions. This study found *PgAP2/ERF* genes were possibly, and importantly, involved in pomegranate fruit development through transcriptomic analysis. Moreover, the *PgAP2/ERF4*, *15*, *26*, *30*, *35*, and *58* genes were screened as crucial candidate genes that might play important roles in pomegranate postharvest preservation based on the *PgAP2/ERF* expression profiles of ‘Tunisian soft-seed’ pomegranate fruit under LTC and 1-MCP + LTC. The detailed regulation mechanisms of pomegranate fruit development and postharvest preservation are to be further investigated.

## Figures and Tables

**Figure 1 genes-13-00895-f001:**
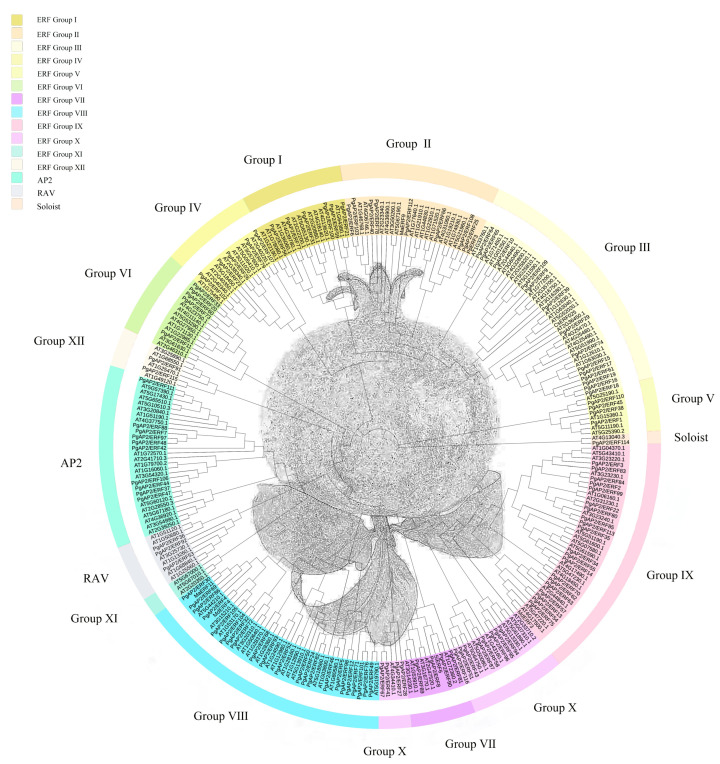
Maximum likelihood phylogeny of the AP2/ERF proteins from pomegranate (116) and *Arabidopsis* (147). The phylogenetic tree was constructed using the MEGA 7.0 tool. Different colors indicate the various subfamilies.

**Figure 2 genes-13-00895-f002:**
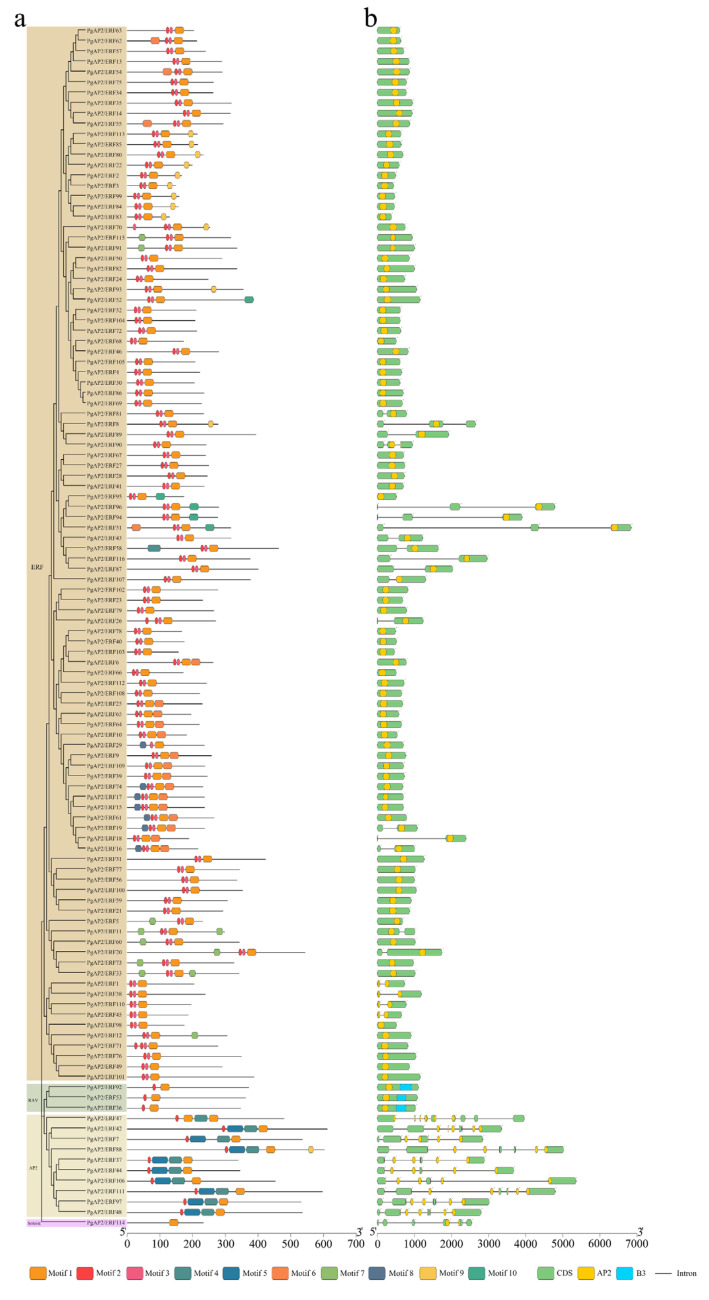
Phylogenetic relationships, gene structure, and conserved protein motifs of the *PgAP2/ERF* genes. The phylogenetic tree was constructed based on the full-length sequences of the PgAP2/ERF proteins using the MEGA 7 tool. (**a**) Motif profile of the PgAP2/ERF proteins. Motifs numbered 1–10 are displayed in differently colored boxes. Sequence information for each motif is presented in Appendix A. (**b**) Exon-intron structures of the *PgAP2/ERF* genes. Protein lengths can be estimated using the scales at the bottom.

**Figure 3 genes-13-00895-f003:**
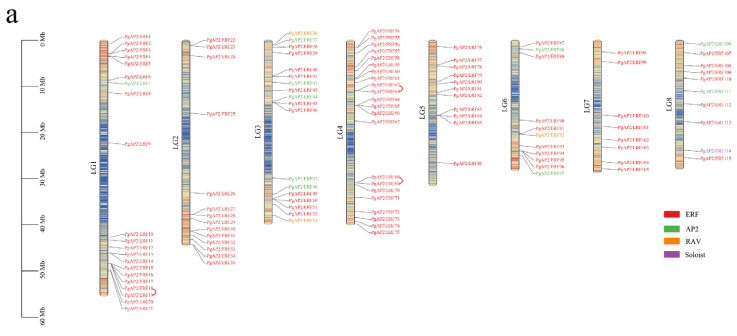
Chromosome distribution and duplication analyses of the *PgAP2/ERF* genes in pomegranate. Vertical bars represent the LG of the pomegranate genome. (**a**) Chromosome distribution of the *PgAP2/ERF* genes. Red and blue represent the level of gene density on the LGs. (**b**) Gene collinearity of the *PgAP2/ERF* gene family. Red lines connecting two chromosomal regions indicate synteny blocks in the pomegranate genome.

**Figure 4 genes-13-00895-f004:**
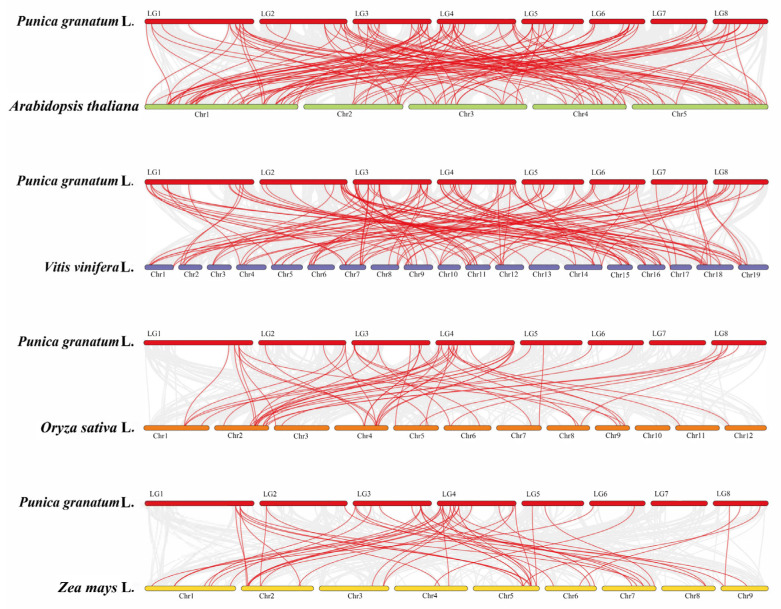
Syntenic analysis of the AP2/ERF genes between pomegranate and *Arabidopsis*, grape, maize, and rice. Gray lines in the background represent the collinear blocks between the different plant genomes, and the red lines indicate the syntenic AP2/ERF gene pairs.

**Figure 5 genes-13-00895-f005:**
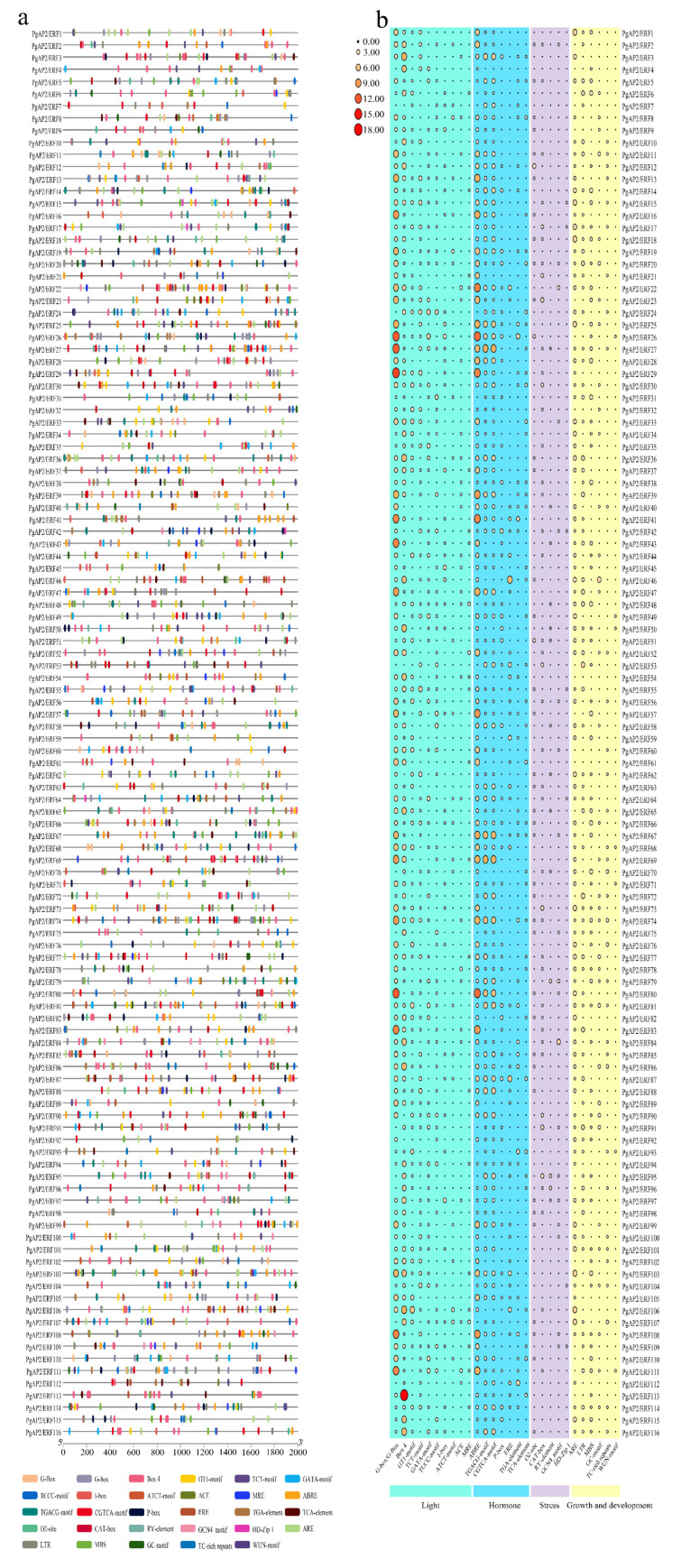
*Cis*-elements of the *PgAP2/ERF* promoters in pomegranate. (**a**) *Cis*-element distribution on the *PgAP2/ERF* promoters. Differentially colored blocks represent the indicated types of *cis*-elements and their locations on the promoters of each *PgAP2/ERF* gene. (**b**) Numbers of the indicated *cis*-elements associated with the indicated biological processes in the promoter regions of each *PgAP2/ERF* gene. Circles with different sizes and different colors from black to yellow and then to red imply that the numbers of the indicated *cis*-elements ranged from 0 to 3 and then to 18.

**Figure 6 genes-13-00895-f006:**
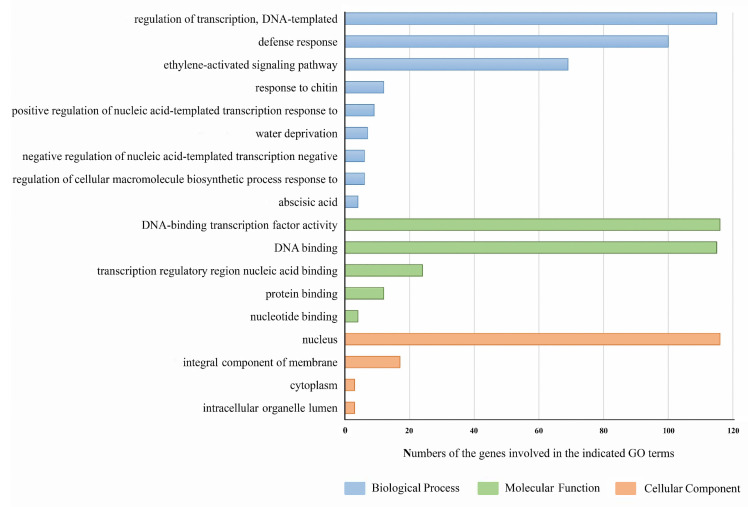
Number of *PgAP2/ERFs* involved in the indicated GO terms of biological processes, molecular functions, and cell component categories according to the Gene Ontology analysis.

**Figure 7 genes-13-00895-f007:**
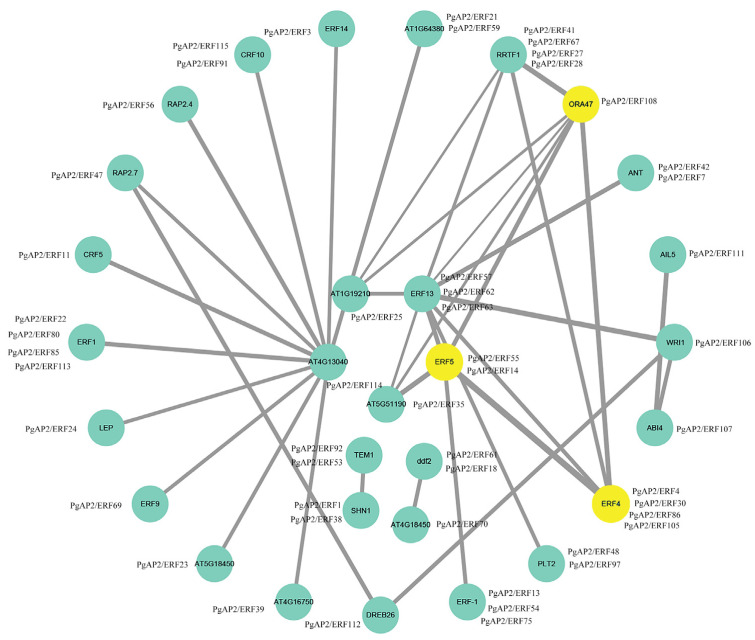
Protein-protein interaction network of PgAP2/ERFs in pomegranate based on AtAP2/ERFs in *Arabidopsis*. PgAP2/ERF proteins are shown by the side of the corresponding Arabidopsis orthologs in the green and yellow circles.

**Figure 8 genes-13-00895-f008:**
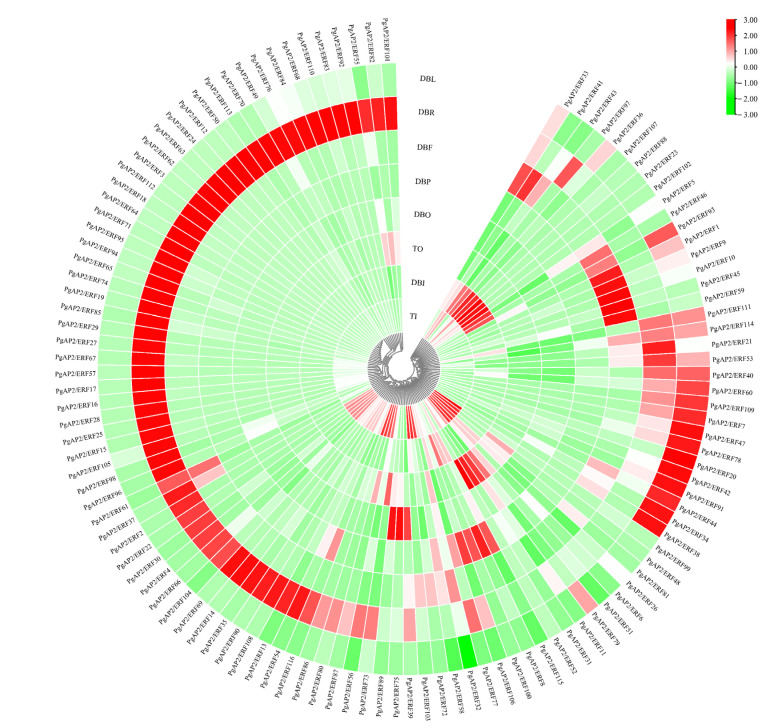
Expression profiles of the *PgAP2/ERF* genes in different pomegranate tissues. DBL, DBR, DBF, DBP, DBO, and DBI: leaf, root, flower, peel, inner, and outer seed coats of ‘Dabenzi’ pomegranate. TI, TO: inner and outer seed coats of ‘Tunisian soft-seed’ pomegranate. Gene expression was calculated as TPM derived from corresponding RNA-Seq data.

**Figure 9 genes-13-00895-f009:**
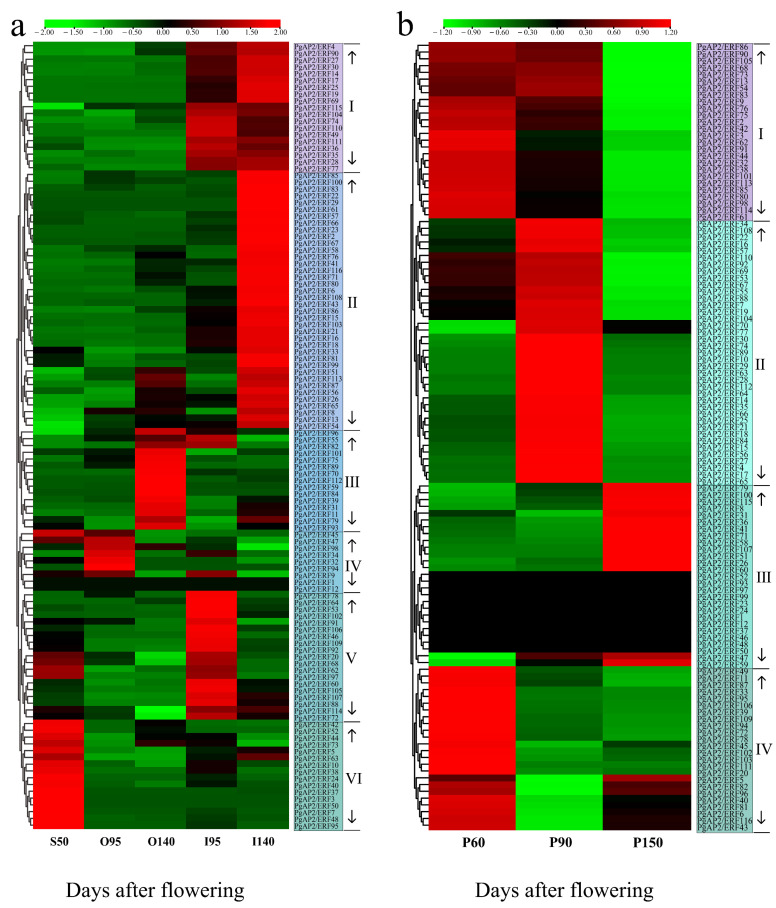
Expression profiles of *PgAP2/ERF* genes during pomegranate fruit development. (**a**) Expression profiles of *PgAP2/ERF* genes in the inner and outer seed coats of ‘Tunisian soft-seed’ pomegranate. S50, O95, O140, I95, and I140: whole seed at 50 days after flowering (DAF), outer seed coats at 95 DAF, outer seed coats at 140 DAF, inner seed coats at 95 DAF, and inner seed coats at 140 DAF, respectively. (**b**) Expression profiles of the *PgAP2/ERF* genes in the peels of ‘Taishanhong’ pomegranate. P60, P90, and P150: peels at 60, 90, and 150 days after flowering, respectively. Gene expression was calculated as TPM derived from corresponding RNA-Seq data.

**Figure 10 genes-13-00895-f010:**
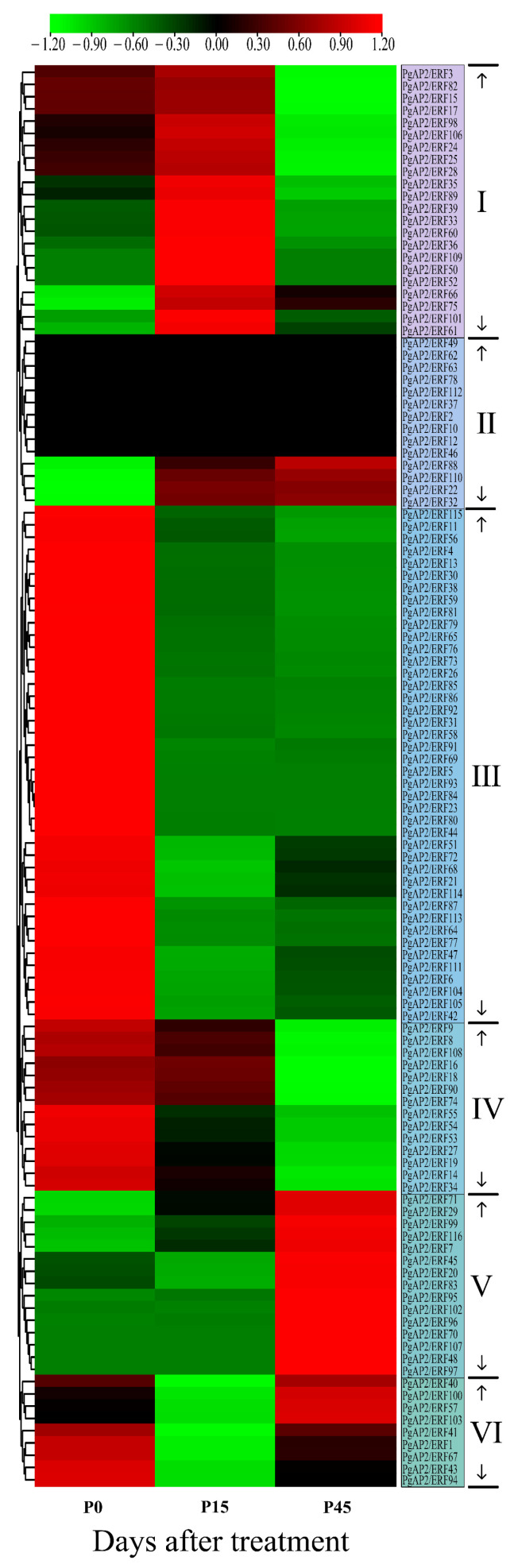
Expression profiles of *PgAP2/ERF* genes during LTC. LTC: cold storage at a low temperature of 4 °C. P0, P15, and P45: ‘Tunisian soft-seed’ pomegranate peels at day 0, 15, and 45 under LTC, respectively. Gene expression was calculated as FPKM derived from corresponding RNA-Seq data.

**Figure 11 genes-13-00895-f011:**
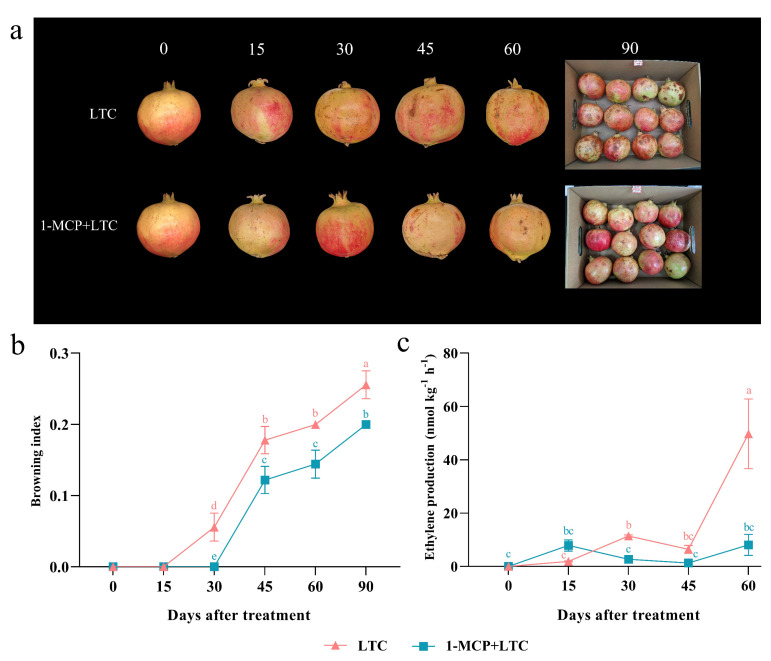
Appearance (**a**), browning index (**b**), and ethylene production level (**c**) of ‘Tunisian soft-seed’ pomegranate fruit under LTC and 1-MCP + LTC at the indicated time points. LTC: cold storage at a low temperature of 4 °C; 1-MCP + LTC: 1-MCP patches under LTC. Data represent the means from three experiments. Error bars represent standard deviations. Different lowercase letters represent significant differences at *p* < 0.05 (Duncan’s test).

**Figure 12 genes-13-00895-f012:**
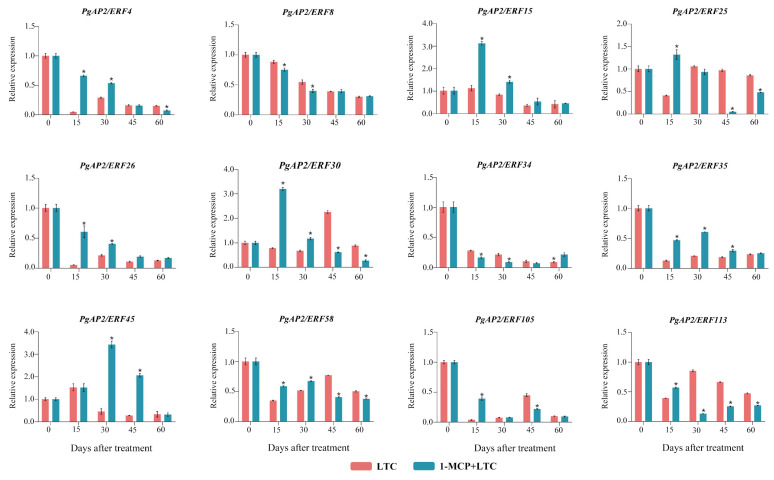
Expression profiles of the selected *PgAP2/ERFs* in ‘Tunisian soft-seed’ pomegranate fruit peels under LTC and 1-MCP + LTC at the indicated time points. LTC: cold storage at a low temperature of 4 °C; 1-MCP + LTC: 1-MCP patches under LTC. Data represent means from three experiments. Error bars represent standard deviations. Asterisks represent significant differences at *p* < 0.05 (*t*-test).

## Data Availability

RNA-seq data (Accession Numbers are presented in Appendix A) from six different pomegranate tissues and organs were retrieved from NCBI Sequence Read Archive data-base (http://www.ncbi.nlm.nih.gov/sra accessed on 15 May 2021).

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
