# Peer review of "Genome-Wide Identification and Comprehensive Analysis of the *AP2/ERF* Gene Family in Pomegranate Fruit Development and Postharvest Preservation"

_genes, 2022, doi:10.3390/genes13050895_

Round 1

Reviewer 1 Report

I have thoroughly reviewed this revised manuscript titled “Genome-wide identification and comprehensive analysis of the AP2/ERF gene family in pomegranate fruit development and postharvest preservation.

The author presents the article nicely with sufficient data.

Reviewer 2 Report

The manuscript shows the identification and analysis of AP2/ERF genes in the pomegranate genome and the effect of cold storage on the expression of these genes. The information in the present manuscript is valuable due to the few antecedents of genes related to phytohormones in this specie. However, some improvements should be done.

Lines 57-67: I see a different font size in this paragraph
Line 82: Please, review a little orthographic mistake.

Line 98: Please, include the 1MCP concentration of patches

Lines 242-245: Please, review that the gene names are written in cursive format.

Lines 492-508: Please, indicate the 1MCP concentration used in the previous studies. The 1MCP concentration can affect the final response.

Lines 516-519: Please, include the scientific name of the fruit species for a better relationship with the gene name.

Lines 536-537: Please, improve the conclusion, e.g. the potential relation of PgAP2/ERF genes with postharvest preservation
Fig 8, 9 and 10: Please, clarify that are results of transcriptome (RNA-seq or other) analysis in the legend.
Fig 11 and 12: Please, include the storage temperature in the legend or figure. As you know the pomegranate susceptibility to ethylene changes depending on storage temperature. Include in the legend that 1MCP was applied as patches.

Reviewer 3 Report

In the submitted manuscript by Qingxia Hu, Yanhui Chen, and colleagues entitled “Genome-wide identification and comprehensive analysis of the AP2/ERF gene family in pomegranate fruit development and postharvest preservation”, the authors examined the effects of AP2/ERF transcription factors in pomegranate development and postharvest preservation. In total, 116 PgAP2/ERF genes in pomegranate were identified and renamed based on their chromosomal distributions. In addition, the phylogenetic relationships with genes from other species were explored. The results of specific TFs associated with the phenotypes of pomegranate seed hardness, fruit development, and cold storage, indicate that these TFs are important candidate genes to use for functional analysis. Overall, the manuscript is well-written, and the figures have a good presentation. The aim and scope of the Genes are in line with the current manuscript.